# $\ell_0$-SPARSE CANONICAL CORRELATION ANALYSIS

**Ofir Lindenbaum**
Faculty of Engineering
Bar Ilan University
Ramat Gan, Israel
ofirlin@gmail.com

**Moshe Salhov**$^*$ **& Amir Averbuch**
School of Computer Science
Tel Aviv University
Tel Aviv, Israel

**Yuval Kluger**
School of Medicine
Yale University
New Haven, CT, USA
yuval.kluger@yale.edu

## ABSTRACT

Canonical Correlation Analysis (CCA) models are powerful for studying the associations between two sets of variables. The canonically correlated representations, termed *canonical variates* are widely used in unsupervised learning to analyze unlabeled multi-modal registered datasets. Despite their success, CCA models may break (or overfit) if the number of variables in either of the modalities exceeds the number of samples. Moreover, often a significant fraction of the variables measures modality-specific information, and thus removing them is beneficial for identifying the *canonically correlated variates*. Here, we propose $\ell_0$-CCA, a method for learning correlated representations based on sparse subsets of variables from two observed modalities. Sparsity is obtained by multiplying the input variables by stochastic gates, whose parameters are learned together with the CCA weights via an $\ell_0$-regularized correlation loss. We further propose $\ell_0$-Deep CCA for solving the problem of non-linear sparse CCA by modeling the correlated representations using deep nets. We demonstrate the efficacy of the method using several synthetic and real examples. Most notably, by gating nuisance input variables, our approach improves the extracted representations compared to other linear, non-linear and sparse CCA-based models.

## 1 INTRODUCTION

Canonical Correlation Analysis (CCA) (Hotelling, 1936; Thompson, 2005), is a classic statistical method for finding the maximally correlated linear transformations of two modalities (or views). Using modalities $\boldsymbol{X} \in \mathbb{R}^{D^x \times N}$ and $\boldsymbol{Y} \in \mathbb{R}^{D^y \times N}$, which are centered and have $N$ samples with $D^x$ and $D^y$ features, respectively, CCA seeks canonical vectors $\boldsymbol{a} \in \mathbb{R}^{D^x}$, and $\boldsymbol{b} \in \mathbb{R}^{D^y}$, such that , $\boldsymbol{u} = \boldsymbol{a}^T \boldsymbol{X}$, and $\boldsymbol{v} = \boldsymbol{b}^T \boldsymbol{Y}$, will maximize the sample correlations between the *canonical variates*, i.e.

$$\max_{\boldsymbol{a}, \, \boldsymbol{b} \neq 0} \quad \rho(\boldsymbol{a}^T \boldsymbol{X}, \boldsymbol{b}^T \boldsymbol{Y}) = \frac{\boldsymbol{a}^T \boldsymbol{X} \boldsymbol{Y}^T \boldsymbol{b}}{\|\boldsymbol{a}^T \boldsymbol{X}\|_2 \|\boldsymbol{b}^T \boldsymbol{Y}\|_2}. \tag{1}$$

To identify non-linear relations between input variables, several extensions of CCA have been proposed. Kernel methods such as Kernel CCA (Bach & Jordan, 2002), Non-parametric CCA (Michaeli et al., 2016) or Multi-view Diffusion maps (Lindenbaum et al., 2020; Salhov et al., 2020) learn the non-linear relations in reproducing Hilbert spaces. These methods have several shortcomings: they are limited to a designed kernel, and they have poor interpolation and extrapolation capabilities. Deep CCA (Andrew et al., 2013) overcomes these limitations by learning two non-linear transformations parametrized using neural networks. The model has several extension, see for example (Wang et al., 2016; Gundersen et al., 2019; Karami & Schuurmans, 2021).

CCA models have been widely used in biology (Pimentel et al., 2018), neuroscience (Al-Shargie et al., 2017), medicine (Zhang et al., 2017), and engineering (Chen et al., 2017), for unsupervised or semi-supervised learning. By extracting meaningful dimensionality reduced representations, CCA improves downstream tasks such as clustering, classification, or manifold learning. One key limitation of CCA is that it requires more samples than features, i.e., $N > D^x, D^y$. However, if we have

---

$^*$M.S. is co affiliated with Playtika Israel

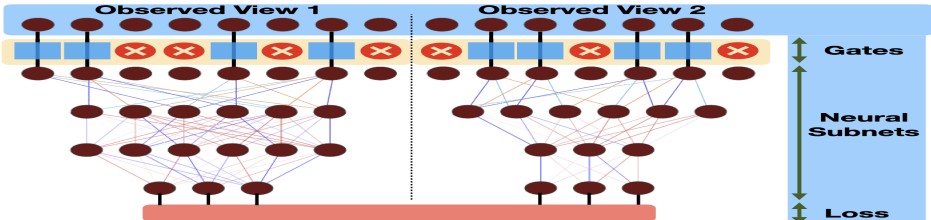

Figure 1: Illustration of $\ell_0$-DCCA. Data from two observed views propagate through stochastic gates (defined in Eq. 4). The gates output is fed into two neural sub-nets that have a shared loss (see Eq. 5). We compute this shared loss based on the neural sub-nets outputs (with dimension $d = 3$ in this example). Our shared loss combines a total correlation term with a differentiable regularization term which induces sparsity in the input variables (by sparsifying the gates).

more variables than samples, the estimation based on the closed-form solution of the CCA problem (in Eq. 1) breaks (Suo et al., 2017) (because the covariance matrix is low rank). Moreover, in high dimensional data, often some of the variables do not measure the phenomenon that is common to both modalities (therefore are not correlated) and thus should be omitted from the transformations. For these reasons, there has been a growing interest in studying sparse CCA models.

Sparse CCA (SCCA) models seek linear transformations which are based on a sparse subset of variables from the input modalities $X$ and $Y$. Sparsifying the feature space improves interpretability, prevents overfitting, and removes the degeneracies inherent to $N < D^x, D^y$ (once the number of selected variables is smaller than $N$ the covariance matrix becomes full rank). To encourage sparsity of the canonical vectors $a$ and $b$, several authors (Wiesel et al., 2008; Cai et al., 2019) propose an $\ell_0$ regularized variant of Eq. 1. However, these schemes are greedy and therefore may lead to suboptimal solutions. As demonstrated by (Waaijenborg et al., 2008; Parkhomenko et al., 2009; Witten et al., 2009; Hardoon & Shawe-Taylor, 2011; Suo et al., 2017) replacing the $\ell_0$ norm by $\ell_1$ is differentiable and leads to a sparse solution to Eq. 1. Several authors extended these ideas by considering group sparsity (Chen & Liu, 2012; Klami et al., 2013; Solari et al., 2019). However, these methods are limited to linear transformations and may lead to shrinkage of the canonical vectors due to the $\ell_1$ regularizer. There has been limited work on extending these models to sparse non-linear CCA. Specifically, there are two kernel-based extensions: two-stage kernel CCA (TSKCCA) by (Yoshida et al., 2017) and SCCA based on Hilbert-Schmidt Independence Criterion (SCCA-HSIC) by (Uurtio et al., 2018). However, these models suffer from the same limitations as KCCA and do not scale to a high dimensional regime.

We present $\ell_0$-CCA, a simple yet effective method for learning correlated representation based on sparse subsets of the input variables by minimizing an $\ell_0$ regularized loss. Our $\ell_0$ regularization relies on a recently proposed Gaussian-based continuous relaxation of Bernoulli random variables, termed gates (Yamada et al., 2020). The gates are applied to the input features to sparsify the canonical vectors. The model and gates parameters are trained jointly via gradient descent to maximize the correlation between the representations of $X$ and $Y$ while simultaneously selecting only the subsets of most correlated input features. Our contributions are four-folds: (i) By modeling the transformations using two neural networks, we provide a natural solution to the problem of sparse non-linear correlation analysis. (ii) We then propose a novel initialization scheme that helps the model identify small subsets of correlated features in the challenging regime of $D > N$. (iii) We apply the proposed method to synthetic data and demonstrate that our approach improves the estimation of the canonical vectors compared with existing baselines. (iv) We demonstrate the efficacy of the proposed scheme to several real work applications; specifically, we demonstrate that it leads to more reliable and interpretable representations than other linear and non-linear data fusion schemes.

## 2 SPARSE CCA

The problem in Eq. 1 has a closed-form solution based on the eigendecomposition of $C_x^{-1} C_{xy} C_y^{-1} C_{yx}$ and $C_y^{-1} C_{yx} C_x^{-1} C_{xy}$, where $C_x, C_y$ are within view sample covariance matrices and $C_{xy}, C_{yx}$ are cross-view sample covariance matrices. However, if $N$ is smaller than the number of input variables ($D^x$ or $D^y$), the sample covariance may be rank deficient, and the closed-form solution becomes meaningless. To overcome this limitation, we consider the problem of sparse CCA. Sparse CCA reduces the number of active variables and therefore the covariance matrix becomes full rank. Sparsity also improves model interpretability but may reduce the maximal (training) correlation value obtained in Eq. 1 (Shalev-Shwartz et al., 2010).

Sparse CCA deals with identifying maximally correlated representations based on linear combinations of sparse subsets of the input variables in $\boldsymbol{X}$ and $\boldsymbol{Y}$. The problem can be formulated as

$$\min_{\boldsymbol{a}, \boldsymbol{b}} \quad -\rho(\boldsymbol{a}^T\boldsymbol{X}, \boldsymbol{b}^T\boldsymbol{Y}) + \lambda^x\|\boldsymbol{a}\|_0 + \lambda^y\|\boldsymbol{b}\|_0, \qquad (2)$$

where $\lambda^x$ and $\lambda^y$ are regularization parameters which control the sparsify the input variables. If $\|\boldsymbol{a}\|_0$ and $\|\boldsymbol{b}\|_0$ are smaller than $N$, we can remove the degeneracy inherent to Eq. 1, and identify meaningful correlated representations based on a sparse subset of input variables.

# 3 PROBABILISTIC REFORMULATION OF SPARSE CCA

The sparse CCA problem formulated in Eq. 2 becomes intractable for large $D^x$ or $D^y$, moreover, due to the discrete nature of the $\ell_0$ norm, the problem is not amenable to gradient-based optimization schemes. Fortunately, as demonstrated in sparse regression, probabilistic models such as the spike-and-slab (George & McCulloch, 1993; Kuo & Mallick, 1998; Zhou et al., 2009; Polson & Sun, 2019) provide a compelling alternative. More recently, differentiable probabilistic models such as (Louizos et al., 2017; Yamada et al., 2020; Jana et al., 2021) were proposed for sparse supervised learning. Here, we adopt these ideas by rewriting the canonical vectors as $\boldsymbol{\alpha} = \boldsymbol{\theta}^x \odot \boldsymbol{s}^x$ and $\boldsymbol{\beta} = \boldsymbol{\theta}^y \odot \boldsymbol{s}^y$, where $\odot$ denotes the Hadamard product (element wise multiplication), and $\boldsymbol{\theta}^x \in \mathbb{R}^{D^x}, \boldsymbol{\theta}^y \in \mathbb{R}^{D^y}$. The vectors $\boldsymbol{s}^x \in \{0,1\}^{D^x}$, $\boldsymbol{s}^y \in \{0,1\}^{D^y}$ are Bernoulli random vectors with parameters $\boldsymbol{\pi}^x = (\pi_1^x, ..., \pi_{D^x}^x)$ and $\boldsymbol{\pi}^y = (\pi_1^y, ..., \pi_{D^y}^y)$. These Bernoulli variables, act as gates and sparsify the coefficients of the canonical vectors. Now, the problem in Eq. 2 can be reformulated as an expectation minimization, which is parameterized by $\boldsymbol{\pi}^x$ and $\boldsymbol{\pi}^y$. Specifically, based on the following theorem, we can reformulate the problem in Eq. 2.

**Theorem 3.1** *The solution of the sparse CCA problem in Eq. 2 is equivalent to the solution of the following probabilistic problem*

$$\min_{\boldsymbol{\pi}^x, \boldsymbol{\pi}^y, \boldsymbol{\theta}^x, \boldsymbol{\theta}^y} \mathbb{E}\big[ -\rho(\boldsymbol{\alpha}^T\boldsymbol{X}, \boldsymbol{\beta}^T\boldsymbol{Y}) + \lambda^x\|\boldsymbol{s}^x\|_0 + \lambda^y\|\boldsymbol{s}^y\|_0 \big], \qquad (3)$$

*where $\boldsymbol{\alpha} = \boldsymbol{\theta}^x \odot \boldsymbol{s}^x$ and $\boldsymbol{\beta} = \boldsymbol{\theta}^y \odot \boldsymbol{s}^y$ and the expectation is taken with respect to the random Bernoulli variables $\boldsymbol{s}^x$ and $\boldsymbol{s}^y$ (which are parameterized by $\boldsymbol{\pi}^x$ and $\boldsymbol{\pi}^y$).*

Note that the expected values of the $\ell_0$ norms can be expressed using the Bernoulli parameters as $\mathbb{E}\|\mathbf{s}^x\|_0 = \sum \pi_i^x$, and $\mathbb{E}\|\mathbf{s}^y\|_0 = \sum \pi_i^y$. The proof relies on the fact that the optimal solution to Eq. 2 is a valid solution to Eq. 3, and vice versa. The proof is presented in the Appendix, Section H, and follows the same construction as the proof of Theorem 1 in (Yin et al., 2020).

## 3.1 CONTINUOUS RELAXATION FOR SPARSE CCA

Due to the discrete nature of $\mathbf{s}^x$ and $\mathbf{s}^y$, differentiating the leading term in Eq. 3 is not straightforward. Although solutions such as REINFORCE (Williams, 1992) or the straight-through estimator (Bengio et al., 2013) enable differentiating through discrete random variables, they still suffer from high variance. Furthermore, they require many Monte Carlo samples for effective training (Tucker et al., 2017). Recently, several authors (Maddison et al., 2016; Jang et al., 2016; Louizos et al., 2017) have demonstrated that using a continuous reparametrization of discrete random variables can reduce the variance of the gradient estimates. Here, following the reparametrization presented in (Yamada et al., 2020; Lindenbaum et al., 2021), we use Gaussian-based relaxation for the Bernoulli random variables.

Each relaxed Bernoulli variables $\mathbf{z}_i$ is defined by drawing from a centered Gaussian $\epsilon_i \sim N(0, \sigma^2)$, then shifting it by $\mu_i$ and truncating its values using the following hard Sigmoid function

$$\mathbf{z}_i = \max(0, \min(1, \mu_i + \epsilon_i)). \qquad (4)$$

Using these relaxed Bernoulli variables, we can define the gated canonical vectors as $\boldsymbol{\alpha} = \boldsymbol{\theta}^x \odot \mathbf{z}^x$ and $\boldsymbol{\beta} = \boldsymbol{\theta}^y \odot \mathbf{z}^y$. We incorporate these vectors into the objective defined in Eq. 3, in which the $\ell_0$ regularization terms can be expressed as

$$\mathbb{E}\|\mathbf{z}^x\|_0 = \sum_{i=1}^{D^x} \mathbb{P}(z_i^x \geq 0) = \sum_{i=1}^{D^x} \left( \frac{1}{2} - \frac{1}{2}\operatorname{erf}\left(-\frac{\mu_i^x}{\sqrt{2}\sigma}\right) \right),$$

where $\mathrm{erf}()$ is the Gaussian error function, and is defined similarly for $\mathbb{E}\|\mathbf{z}^y\|_0$.

To learn to model parameters $\boldsymbol{\theta}^x, \boldsymbol{\theta}^y$ and gate parameters $\boldsymbol{\mu}^x, \boldsymbol{\mu}^y$ we first draw realizations for the gates, then we update the parameters by applying gradient descent to minimize

$$\mathbb{E}\big[-\rho(\boldsymbol{\alpha}^T\boldsymbol{X}, \boldsymbol{\beta}^T\boldsymbol{Y}) + \lambda^x\|\mathbf{z}^x\|_0 + \lambda^y\|\mathbf{z}^y\|_0\big].$$

Post training, we remove the stochasicity from the gates and use all variables such that $z_i^x = \max(0, \min(1, \mu_i^x)) > 0$ (defined similarly for $\mathbf{z}^y$).

## 3.2 Gate Initialization

If the gates are initialized with $\mu_i = 0.5$, they will approximate "fair" Bernoulli variables. This is a reasonable choice if no prior knowledge about the solution is available; however, we can utilize the closed-form solution of the CCA problem to derive a more suitable parameter initialization for the gate. Specifically, given the empirical covariance matrix $\boldsymbol{C}_{xy} = \frac{\boldsymbol{X}\boldsymbol{Y}^T}{(N-1)}$, we denote the thresholded covariance matrix by $\bar{\boldsymbol{C}}_{xy}$, with values defined as follows

$$(\bar{C}_{xy})_{ij} = \begin{cases} (C_{xy})_{i,j}, & \text{if } |(C_{xy})_{i,j}| > \delta \\ 0, & \text{otherwise,} \end{cases}$$

where $\delta$ is the selected threshold value based on the desired sparsity for $\boldsymbol{X}$ and $\boldsymbol{Y}$, specifically, if we assume that $r$ percent of the values should be zeroed, then $\delta$ is set to be the $r$-th percentile of $|(C_{xy})|$. Then, we compute the leading singular vectors $\boldsymbol{u}$ and $\boldsymbol{v}$ of $\bar{\boldsymbol{C}}_{xy}$. We further threshold the absolute values of these vectors using the same percentile used for $\bar{\boldsymbol{C}}_{xy}$. The initial values of the parameters of the gates are then defined by $\boldsymbol{\mu}^x = \bar{\boldsymbol{u}} + 0.5$, and $\boldsymbol{\mu}^y = \bar{\boldsymbol{v}} + 0.5$, where $\bar{\boldsymbol{u}}$ and $\bar{\boldsymbol{v}}$ are the thresholded versions of the absolute value of the singular vectors. Using this procedure, we increase the initial probability for all gates in the support of $\bar{\boldsymbol{u}}$ and $\bar{\boldsymbol{v}}$ based on the singular vectors of $\bar{\boldsymbol{C}}_{xy}$.

## 3.3 $\ell_0$-Deep CCA

To extend our model to non-linear function estimation, we can formulate the problem of sparse non-linear CCA by modeling the transformations using deep nets as in (Andrew et al., 2013; Wang et al., 2015b). We introduce two random Bernoulli relaxed vectors into the input layers of two neural networks trained in tandem to maximize the total correlation. Denoting the random gating vectors $\mathbf{z}^x$ and $\mathbf{z}^y$ for view $\boldsymbol{X}$ and $\boldsymbol{Y}$, respectively, our $\ell_0$-Deep CCA ($\ell_0$-DCCA) loss is defined by

$$\mathbb{E}\big[-\bar{\rho}(\boldsymbol{f}(\hat{\boldsymbol{X}}), \boldsymbol{g}(\hat{\boldsymbol{Y}})) + \lambda^x\|\mathbf{z}^x\|_0 + \lambda^y\|\mathbf{z}^y\|_0\big], \tag{5}$$

where $\boldsymbol{f}(\hat{\boldsymbol{X}}) = \boldsymbol{f}(\mathbf{z}^x \odot \boldsymbol{X}|\boldsymbol{\theta}^x) \in \mathbb{R}^{d \times N}$, and $\boldsymbol{g}(\hat{\boldsymbol{Y}}) = \boldsymbol{g}(\mathbf{z}^y \odot \boldsymbol{Y}|\boldsymbol{\theta}^y) \in \mathbb{R}^{d \times N}$ are modeled as deep networks with model parameters $\boldsymbol{\theta} = (\boldsymbol{\theta}^x, \boldsymbol{\theta}^y)$, and gate parameters $\boldsymbol{\mu} = (\boldsymbol{\mu}^x, \boldsymbol{\mu}^y)$. The functions $\boldsymbol{f}$ and $\boldsymbol{g}$ embed the data into a $d$-dimensional space (this formulation could also be used for linear $\boldsymbol{f}$ and $\boldsymbol{g}$). The functional $\bar{\rho}(\boldsymbol{f}(\hat{\boldsymbol{X}}), \boldsymbol{g}(\hat{\boldsymbol{Y}}))$ measures the total correlation between the two $d$ dimensional outputs of the deep nets, this is the sum over $d$ correlation values computed between pairs of coordinates. Exact details on the computation of this term appear in the following subsection. The regularization parameters $\lambda^x, \lambda^y$ control the sparsity of the input variables. The vectors $\mathbf{z}^x$ and $\mathbf{z}^y$ are Bernoulli relaxed vectors, with elements defined based on Eq. 4.

Figure. 1 highlights the proposed architecture. We first pass both observed modalities through the gates. Then, we feed these into view-specific neural sub-nets. Finally, we minimize the shared loss term in Eq. 5 by optimizing the parameters of the gates and the neural sub-nets.

## 3.4 Algorithm details

Denoting the centered representations for $\boldsymbol{X}, \boldsymbol{Y}$ by $\boldsymbol{\Psi}^x, \boldsymbol{\Psi}^y \in \mathbb{R}^{d \times N}$ (computed using the coupled neural sub-nets), respectively, the empirical covariance matrix between these representations can be expressed as $\widehat{\boldsymbol{C}}_{xy} = \frac{1}{N-1}\boldsymbol{\Psi}^x(\boldsymbol{\Psi}^y)^T$. Using a similar notations, we express the regularized empirical covariance matrices of $\boldsymbol{X}$ and $\boldsymbol{Y}$ as $\widehat{\boldsymbol{C}}_x = \frac{1}{N-1}\boldsymbol{\Psi}^x(\boldsymbol{\Psi}^x)^T + \gamma\boldsymbol{I}$ and $\widehat{\boldsymbol{C}}_y = \frac{1}{N-1}\boldsymbol{\Psi}^y(\boldsymbol{\Psi}^y)^T + \gamma\boldsymbol{I}$,

where the matrix $\gamma \boldsymbol{I}$ ($\gamma > 0$) is added to stabilize the invertibility of $\widehat{C}_x$ and $\widehat{C}_y$. The total correlation in Eq. 5 (i.e. $\bar{\rho}(\boldsymbol{f}(\hat{X}), \boldsymbol{g}(\hat{Y}))$) can be expressed using the trace of $\widehat{C}_y^{-1/2} \widehat{C}_{yx} \widehat{C}_x^{-1} \widehat{C}_{xy} \widehat{C}_y^{-1/2}$.

To learn the parameters of the gates $\boldsymbol{\mu}$ and of the representations $\boldsymbol{\theta}$ we apply full batch gradient decent to the loss in Eq. 5. Specifically, we use Monte Carlo sampling to estimate the left part of Eq. 5. This is repeated for several steps (epochs), using one Monte Carlo sample between each gradient step as suggested by (Louizos et al., 2017) and (Yamada et al., 2020), and worked well in our experiments. After training, we remove the stochastic part of the gates and use only the variables $i^x \in \{1, ..., D^x\}$ and $i^y \in \{1, ..., D^y\}$ such that $z_{i^x}^x > 0$ and $z_{i^y}^y > 0$. Finally, we apply linear CCA to the low dimensional extracted embedding to remove this arbitrary scaling and use the normalized generalized eigenvectors (of the closed-form CCA solution) as our canonical vectors. Empirically, we observe that our method also works well with stochastic gradient descent, as long as the batch size is not too small. Alternatively, for small batches we can use a variant of the total correlation loss as was presented in (Wang et al., 2015c) or (Chang et al., 2018). In the Appendix section G, we present a pseudo-code of $\ell_0$-DCCA, evaluate the complexity of the method, and extend it for a multi-modal setting (more than two views).

## 4 RELATED WORK

The problem of of $\ell_0$ sparse CCA was studied in (Wiesel et al., 2008; Cai et al., 2019). However, both rely on a greedy heuristic which iteratively adds non zero elements to the support of $\boldsymbol{a}$ and $\boldsymbol{b}$. Alternatively, as proposed by (Suo et al., 2017), an $\ell_1$-regularized problem can be described as

$$\boldsymbol{a}, \boldsymbol{b} = \operatorname{argmin} \left[ - \operatorname{Cov}(\boldsymbol{a}^T \boldsymbol{X}, \boldsymbol{b}^T \boldsymbol{Y}) + \tau_1 \|\boldsymbol{a}\|_1 + \tau_2 \|\boldsymbol{b}\|_1 \right],$$
$$\text{subject to} \quad \|\boldsymbol{a}^T \boldsymbol{X}\|_2 \leq 1, \quad \|\boldsymbol{b}^T \boldsymbol{Y}\|_2 \leq 1,$$

where $\tau_1$ and $\tau_2$ are regularization parameters. A number of variants have been proposed to this problem (Waaijenborg et al., 2008; Parkhomenko et al., 2009; Witten et al., 2009; Hardoon & Shawe-Taylor, 2011), but they all suffer from parameter shrinkage and therefore lead to a solution which is less consistent with the correct canonical vectors (see Table 1).

## 5 EXPERIMENTAL RESULTS

We validate the effectiveness of the proposed approach on a wide range of tasks. First, using synthetic data, we demonstrate that $\ell_0$-CCA correctly identifies the canonical vectors in a challenging regime of $N < D^x, D^y$. Next, using a coupled video dataset, we demonstrate that $\ell_0$-CCA can identify the common information from high dimensional data, and embed it into correlated, low-dimensional representations. Then, we use noisy images from MNIST and multi-channel seismic data to demonstrate that $\ell_0$-DCCA finds meaningful representations of the data even in a noisy regime. Finally, we use $\ell_0$-DCCA to improve cancer sub-type classification using high dimensional genetic measurements. In all experiments validation sets are used for tuning the hyperparameters of all baselines by maximizing the total correlation on the validation set. We refer the reader to the Appendix for a complete description of the baselines, training procedure, and parameters choice for all methods.

### 5.1 SYNTHETIC EXAMPLE

To generate samples from $\boldsymbol{X}, \boldsymbol{Y} \in \mathbb{R}^{D \times N}$, we follow the procedure described in (Suo et al., 2017), considering data sampling from the following distribution $\begin{pmatrix} \boldsymbol{X} \\ \boldsymbol{Y} \end{pmatrix} \sim N(\begin{pmatrix} \boldsymbol{0} \\ \boldsymbol{0} \end{pmatrix}, \begin{pmatrix} \boldsymbol{\Sigma}_x & \boldsymbol{\Sigma}_{xy} \\ \boldsymbol{\Sigma}_{yx} & \boldsymbol{\Sigma}_y \end{pmatrix})$, where $\boldsymbol{\Sigma}_{xy} = \rho_0 \boldsymbol{\Sigma}_x (\boldsymbol{\phi}\boldsymbol{\eta}^T) \boldsymbol{\Sigma}_y$. We study three cases for the covariance matrices $\boldsymbol{\Sigma} = \boldsymbol{\Sigma}_x = \boldsymbol{\Sigma}_y$.
**Model I.** Identity: $\boldsymbol{\Sigma} = \boldsymbol{I}_D$. **Model II.** Toeplitz: $\Sigma_{i,j} = \rho_0^{|i-j|}, i, j = 1, ..., D$.
**Model III.** Sparse inverse: $\Sigma_{i,j} = \frac{\bar{\Sigma}_{i,j}}{\sqrt{\bar{\Sigma}_{i,i} \bar{\Sigma}_{j,j}}}$, where $\bar{\boldsymbol{\Sigma}} = \boldsymbol{\Gamma}^{-1}, \Gamma_{i,j} = \mathbb{1}_{i=j} + 0.5\mathbb{1}_{i=j} + 0.4\mathbb{1}_{i=j}$.

For all three cases, the vectors $\boldsymbol{\phi}, \boldsymbol{\eta} \in \mathbb{R}^D$, are sparse with 5 nonzero elements and $\rho_0 = 0.9$. The indices of the active elements are chosen randomly with values equal to $1/\sqrt{5}$. In this setting, the

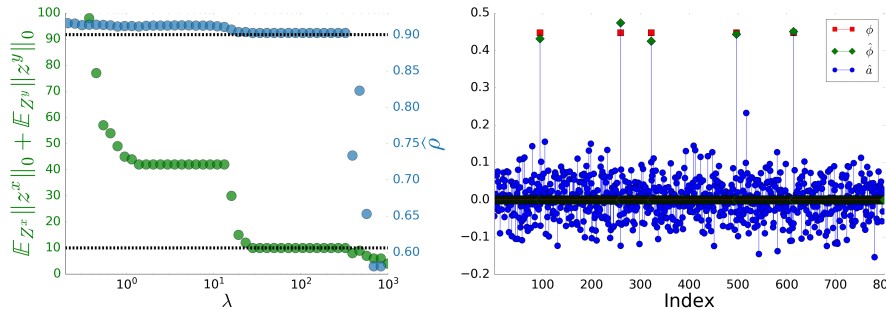

Figure 2: Left: Regularization path of $\ell_0$-CCA on data generated from the linear model I (described in Section 5.1). Values on the left $y$-axis (green) represent the sum of active gates (by expectation). Values on the right $y$-axis (blue) represent the empirical correlation between the estimated representations, i.e. $\hat{\rho} = \hat{\phi}^T X^T Y \hat{\eta}$, where $\hat{\phi}$ and $\hat{\eta}$ are the estimated canonical vectors. Dashed lines indicate the correct number of active coefficients (10) and true correlation $\rho$ (0.9). Note that for small values of $\lambda = \lambda^x = \lambda^y$, the model selects many variables and attains a higher correlation value; in this case, $\ell_0$-CCA suffers from overfitting. Right: True canonical vector $\phi$ along with the estimated vectors using $\ell_0$-CCA ($\hat{\phi}$) and CCA ($\hat{a}$). Due to the small sample size, CCA overfits and fails to identify the correct canonical vectors.

canonical vectors $a$ and $b$ maximizing the correlation objective in Eq. 1 are $\phi$ and $\eta$, respectively (see Proposition 1 in (Suo et al., 2017)).

| | Model I- Identity covariance | | |
|---|---|---|---|
| $(N, D^x, D^y)$ | (400,800,800) | (500,600,600) | (700,1200,1200) |
| PMA (Witten et al., 2009) | (1.170,1.170) | (0.850,0.850) | (1.090,1.090) |
| IP-SCCA (Mai & Zhang, 2019) | (1.658,1.647) | (1.051,1.051) | (1.544,1.542) |
| SCCA-I (Hardoon & Shawe-Taylor, 2011) | (1.602,1.140) | (1.143,0.282) | (1.160,0.181) |
| SCCA-II (Gao et al., 2017) | (0.060,0.066) | (0.053,0.057) | (0.045,0.043) |
| mod-SCCA (Suo et al., 2017) | (0.056,0.062) | (0.05,0.056) | (0.045,0.043) |
| $\ell_0$-CCA | **(0.003,0.009)** | **(0.002,0.002)** | **(0.001,0.002)** |
| | Model II- Toeplitz covariance | | |
| PMA (Witten et al., 2009) | (1.038,1.067) | (1.115,0.943) | (1.098,0.890) |
| IP-SCCA (Mai & Zhang, 2019) | (NA,NA) | (NA,NA) | (NA,NA) |
| SCCA-I (Hardoon & Shawe-Taylor, 2011) | (1.382,1.357) | (1.351,1.299) | (1.219,1.186) |
| SCCA-II (Gao et al., 2017) | (0.213,0.296) | (0.145,0.109) | (0.110,0.088) |
| mod-SCCA (Suo et al., 2017) | (0.173,0.218) | (0.136,0.098) | (0.109,0.086) |
| $\ell_0$-CCA | **(0.101,0.079)** | **(0.098,0.072)** | **(0.026,0.039)** |
| | Model III- Sparse Inverse covariance | | |
| PMA (Witten et al., 2009) | (0.930,1.050) | (0.670,0.450) | (0.760,0.580) |
| IP-SCCA (Mai & Zhang, 2019) | (0.654,0.653) | (0.092,0.091) | (0.282,0.285) |
| SCCA-I (Hardoon & Shawe-Taylor, 2011) | (1.375,0.966) | (1.041,0.502) | (0.985,0.364) |
| SCCA-II (Gao et al., 2017) | (0.129,0.190) | (0.069,0.062) | (0.051,0.047) |
| mod-SCCA (Suo et al., 2017) | **(0.092,**0.149) | (0.068,0.059) | (0.050,0.044) |
| $\ell_0$-CCA | (0.108,**0.103)** | **(0.026,0.036)** | **(0.009,0.005)** |

Table 1: Evaluating the estimation quality of the canonical vectors $\phi$ and $\eta$. Each pair indicates $(e_\phi, e_\eta)$, which are the estimation errors of $\phi$ and $\eta$ respectively. We compare the proposed $\ell_0$-CCA to other sparse CCA schemes considering three types of covariance matrices for generating $X$ and $Y$, and different dimensions $(N, D^x, D^y)$. The description of all three covariances appears in Section 5.1. We highlight the smallest error obtained across all methods using boldface.

Using Model I, we first generate $N = 400$ samples, with $D = 800$, and estimate the canonical vectors based on CCA and $\ell_0$-CCA. In Fig. 2, we present a regularization path of the proposed scheme. Specifically, we apply $\ell_0$-CCA to the data described above using various values of $\lambda = \lambda^x = \lambda^y$. We present the $\ell_0$ of active gates (by expectation) along with the empirical correlation between the extracted representations, defined by $\hat{\rho} = \rho(\hat{\phi}^T X, \hat{\eta}^T Y)$. As evident from the left panel, a wide range of $\lambda$ values leads to the correct number of active coefficients (10) and the correct correlation value ($\rho_0 = 0.9$). Next, in the right panel we present the values of $\phi$, the $\ell_0$-CCA estimate (using $\lambda = 30$) of the canonical vector $\hat{\phi}$, and the CCA spectral based estimate of the canonical vector $\hat{a}$. Due to the low number of samples, the solution by CCA is wrong and not sparse, while the $\ell_0$-CCA solution correctly identifies the support of $\phi$.

Next, we evaluate the estimation error of $\phi$ using $e_{\boldsymbol{\phi}} = 2(1 - |\boldsymbol{\phi}^T \hat{\boldsymbol{\phi}}|)$, and $e_{\boldsymbol{\eta}}$ is defined similarly. In Table 1 we present the estimation errors of $\phi$ and $\rho$ (averaged over 100 simulations) for Models I, II and III (identity, Toeplitz and sparse inverse covariance matrices). As baselines, we compare the performance to 5 leading sparse CCA models. As evident from these experiments, $\ell_0$-CCA significantly outperforms all baselines in its ability to learn the correct canonical vectors. Here, we restrict our evaluation to five baselines since code was not available for other sparse CCA models. We have implemented (Wiesel et al., 2008) but observed that it does not converge to the correct canonical vectors (the estimation errors $e_{\boldsymbol{\phi}}$ and $e_{\boldsymbol{\rho}}$ were 2). In the Appendix, section F we provide a runtime evaluation of the method for different values of $N$ and $D$.

## 5.2 MULTI VIEW OF SPINNING PUPPETS

As an illustrative example, we use a dataset collected by (Lederman & Talmon, 2018). The authors have generated two videos capturing rotations of 3 desk puppets. One camera captures two puppets, while the other captures another two, where one puppet is shared across cameras. A snapshot from both cameras appears in Fig. 3. All puppets are placed on spinning devices that rotate the dolls at different frequencies. In both videos, there is a shared underlying parameter, namely the rotation of the common bulldog. We use 400 images from each camera, where each image has $240 \times 320 = 76800$ pixels (using a grayscaled version of the colored image); therefore, $N \ll D$, and direct application of CCA would fail. We apply the proposed scheme using $\lambda^y = \lambda^x = 50$, a linear activation and embedding dimension $d = 2$. $\ell_0$-CCA converges to embedding with a total correlation of 1.99 using 372 and 403 pixels from views $\boldsymbol{X}$ and $\boldsymbol{Y}$. The active gates are presented in the right panels of Fig. 3. In this example, the gates correctly attenuate the modality-specific information and preserve subsets of pixels capturing common information, allowing for embedding the gated videos into a shared correlated representation.

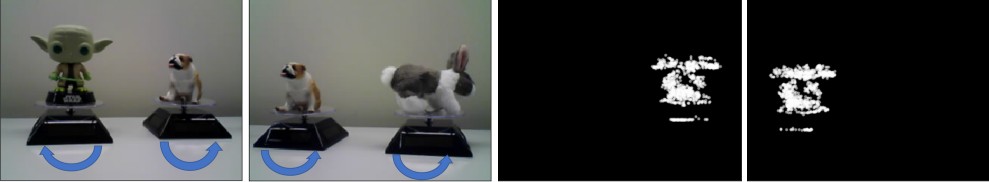

Figure 3: Left: two samples from the spinning puppets videos. The videos capture the rotation of 3 desk puppets. Arrows indicate the spinning direction of each puppet. We use $\ell_0$-CCA to identify a sparse subset of pixels that are correlated from both videos. Right: the values of the gates for each video $\mathbf{z}^x$ and $\mathbf{z}^y$. After training, the values of the gates are binary (i.e., $\{0, 1\}$), with 372 and 403 active gates for the left and right videos, respectively. $\ell_0$-CCA correctly select correlated subsets of pixels that highlight the common puppet (Bulldog) in this example.

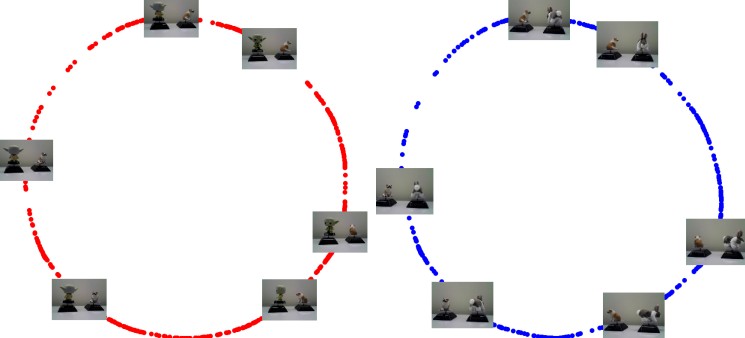

Figure 4: The correlated $\ell_0$-CCA representations (visualized using MVDM (Lindenbaum et al., 2020)) of the Yoda+Bulldog video (left) and Bulldog+Bunny (right). We superimpose each embedding with 6 images corresponding to 6 points in the embedding spaces. The transformations are based on the information described by pixels whose gates are active (presented in Fig. 3). The resulting embeddings are correlated with each other, with a total correlation of $\bar{\rho} = 1.99$. The structure captured by the embeddings correctly represent the angular rotation of the Bulldog, which is the common latent parameter in this experiment.

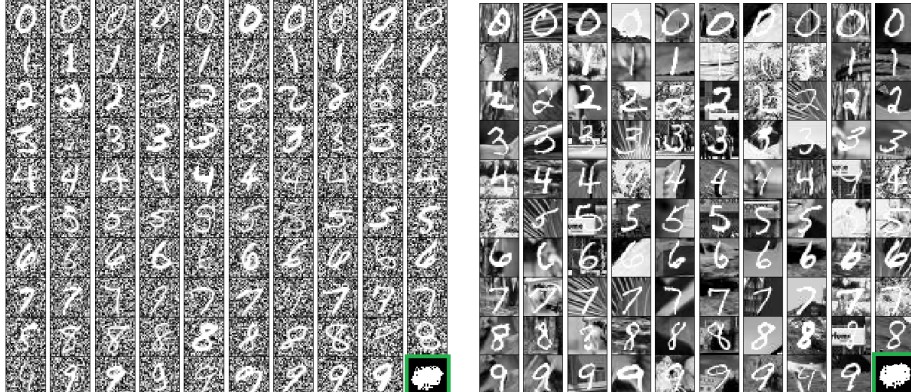

Figure 5: Images from the coupled noisy MNIST dataset. In the bottom right of both panels, we presents the active gates (white values within a green frame). There are 277 and 258 active gates for view I and II, respectively.

In Fig. 4, we present the coupled two-dimensional embedding of both videos. Both embeddings are correlated with each other and reveal the correct angular orientation of the Bulldog (which is the common latent parameter in this experiment). This indicates the ability of $\ell_0$-CCA to identify correlated latent variables in the regime of $N \ll D^x, D^y$.

### 5.3 NOISY MNIST

We use two noisy variants of MNIST (LeCun et al., 2010) as our coupled views. The first view is created by adding noise drawn uniformly from $[0, 1]$ to all pixels. The second view is created by placing a random patch from a natural image in the background of the handwritten digits. Random samples from these modalities are presented in Fig. 5. Each view consists of $62,000$ samples, of which we use $40,000$ for training, $12,000$ for testing and $10,000$ are used as a validation set.

We train $\ell_0$-DCCA to embed the data into a correlated 10 dimensional space while selecting subsets of input pixels. Our model selects 277, and 258 pixels from both modalities respectively (see bottom right corner of Fig. 5). Next, we evaluate the quality of learned embedding by applying $k$-means to the stacked embedding of both views. We run $k$-means (with $k = 10$) using 20 random initializations and record the run with the smallest sum of square distances from centroids. Given the cluster assignment, $k$-means clustering accuracy (KM) and mutual information (MI) are measured using the true labels. Additionally, we train a Linear-SVM (SVM) model on our train and validation datasets. SVM classification accuracy is measured on the remaining test set. The embedding provided by $\ell_0$-DCCA leads to higher classification and clustering results compared with several linear and non-linear modality fusion models appear in Table 2. In the Appendix, we provide the implementation details and present experimental results demonstrating the accuracy vs. sparsity trade-off of $\ell_0$-DCCA.

### 5.4 SEISMIC EVENT CLASSIFICATION

Next, we evaluate the method using a dataset of seismic events studied by (Lindenbaum et al., 2018). Here, we focus on 537 explosions which are categorized into 3 quarries. Each event is recorded using two directional channels facing east (E) and north (N); these comprise the coupled views for the correlation analysis. Following the analysis by (Lindenbaum et al., 2018), the input features are sonogram representations of the seismic signal. Sonograms are time-frequency representations with bins equally tempered on a logarithmic scale. Each sonogram $z \in \mathbb{R}^{1157}$ with 89 time bins and 13 frequency bins. An example of sonograms from both channels appears in the top row of Fig. 6.

We create the noisy seismic data by adding sonograms computed based on vehicle noise from [1]. Examples of noisy sonograms appear in the middle row of Fig. 6. We hold out $20\%$ of the data as a validation set, and train $\ell_0$-DCCA to embed the data in 3 dimensions. In Table 2 we present the MI, $k$-means and SVM accuracies computed based on $\ell_0$-DCCA embedding. Furthermore, we compare the performance with several other baselines. Here, the proposed scheme improves performance in all 3 metrics while identifying a subset of 17 and 16 features from channel E and N, respectively. The active gates are presented in the bottom row of Fig. 6. Our results indicate that even in the presence of strong noise, $\ell_0$-DCCA correctly activates the gates in frequency bins that coincide with the energy stamps of the primary and secondary waves (P and S in the top left of Fig. 6).

---

[1] https://bigsoundbank.com/search?q=car

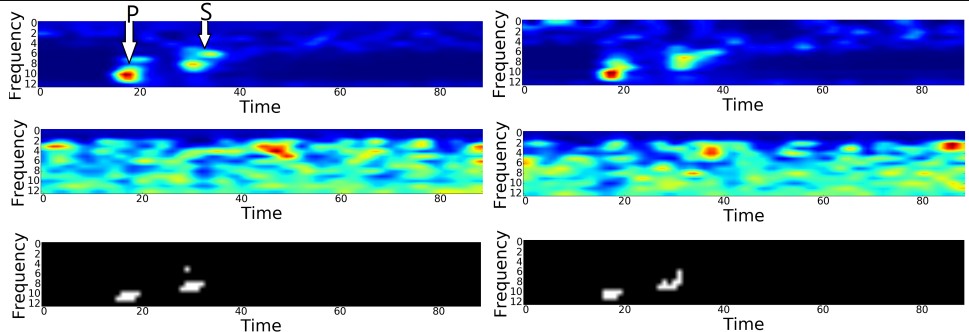

Figure 6: Top: Clean sample sonograms of an explosion based on the E and N channels (left and right, respectively). Arrows highlight the Primary (P) and Secondary (S) waves caused by the explosion. Middle: noisy sonograms generated by adding sonograms of vehicle recordings. Bottom: the active gates for both channels. Note that the gates are active at time-frequency bins which correspond to the P and S waves (see top left panel).

| Method | Noisy MNIST (LeCun et al., 2010) | | | Seismic (Lindenbaum et al., 2018) | | | METABRIC (Curtis et al., 2012) | | |
|---|---|---|---|---|---|---|---|---|---|
| | MI | KM (%) | SVM (%) | MI | KM (%) | SVM (%) | MI | KM (%) | SVM (%) |
| Raw Data | 0.130 | 16.6 | 86.6 | 0.001 | 35.7 | 41.3 | 0.58 | 36.5 | 63.8 |
| PCA (Pearson, 1901) | 0.130 | 16.6 | 89.3 | 0.002 | 38.8 | 41.3 | 0.08 | 19.2 | 23.6 |
| CCA (Chaudhuri et al., 2009) | 1.290 | 66.4 | 75.8 | 0.003 | 38.1 | 40.4 | 0.20 | 20.7 | 24.1 |
| mod-SCCA (Suo et al., 2017) | 0.342 | 23.9 | 63.1 | 0.610 | 71.7 | 86.9 | 0.12 | 21.0 | 26.0 |
| SCCA-HSIC (Uurtio et al., 2018) | NA | NA | NA | 0.003 | 38.7 | 49.5 | NA | NA | NA |
| KCCA (Bach & Jordan, 2002) | 0.943 | 50.2 | 85.3 | 0.006 | 38.4 | 92.5 | 0.35 | 30.8 | 61.3 |
| grad-KCCA (Uurtio et al., 2019) | NA | NA | NA | 0.005 | 40.9 | 41.4 | 0.50 | 32.6 | 47.8 |
| multiview-ICA (Richard et al., 2020) | 1.750 | 88.0 | 90.0 | 0.748 | 90.1 | 94.2 | 0.74 | 44.7 | 62.8 |
| NCCA (Michaeli et al., 2016) | 1.030 | 47.5 | 77.2 | 0.700 | 86.8 | 91.4 | 0.72 | 48.7 | 63.7 |
| DCCA (Andrew et al., 2013) | 1.970 | 93.2 | 93.2 | 0.830 | 94.9 | 94.6 | 0.79 | 45.2 | 72.1 |
| DCCAE (Wang et al., 2015b) | 1.940 | 91.8 | 94.0 | 0.92 | 97.0 | 97.0 | 0.68 | 42.9 | 69.0 |
| $\ell_0$-CCA (linear) | 1.73 | 87.1 | 88.4 | 0.85 | 93.7 | 94.9 | 0.82 | 49.5 | 70.7 |
| $\ell_0$-DCCA (non-linear) | **2.05** | **95.4** | **95.5** | **0.97** | **98.1** | **97.2** | **0.88** | **50.3** | **74.1** |

Table 2: Evaluation of correlated embedding extracted from the Noisy MNIST, Seismic, and METABRIC (cancer type) datasets. The representation extracted by $\ell_0$-DCCA leads to higher clustering accuracy (KM), and classification accuracy (SVM) compared with several baselines. MI indicates the Mutual Information between cluster assignments and sample labels. We use NA to denote simulations which did not converge.

## 5.5 CANCER SUB-TYPE CLASSIFICATION

Accurate classification of cancer sub-type is vital for extending the life span of patients by personalized treatments (Zhu et al., 2020). This task is challenging since the number of measured genes is typically much larger than the number of observations. Here, we use multi-modal observations from the METABRIC data (Curtis et al., 2012) and attempt to find correlated representations to improve cancer sub-type classification. The data consists of $1,112$ breast cancer patients which are annotated by 10 subtypes based on InClust (Dawson et al., 2013). We observe two modalities, namely the RNA gene expression data, and Copy Number Alteration (CNA) data. The dimensions of these modalities are $15,709$ and $47,127$, respectively. We compute the $\ell_0$-DCCA 10 dimensional embedding (and all baseline embeddings) and demonstrate using $k$-means and SVM that the representation identified using $\ell_0$-DCCA can lead to more accurate cancer sub-type classification (see Table 2).

## 6 CONCLUSION

This paper presents a method for learning sparse non-linear transformations that maximize the canonical correlations between two modalities. Our approach is realized by gating the input layers of two neural networks, which are trained to optimize their output's total correlations. Input variables are gated using a regularization term which encourages sparsity. We further propose a novel scheme to initialize the gates based on a thresholded cross-covariance matrix. Our method can learn informative correlated representations even when the number of variables far exceeds the number of samples. Finally, we demonstrate that the proposed scheme outperforms existing algorithms for linear and non-linear canonical correlation analysis. We describe our future directions in the Appendix section I.

## 7 ACKNOWLEDGMENT

The work of YK was supported by the National Institutes of Health R01GM131642, UM1PA05141, U54AG076043, P50CA121974, and U01DA053628.

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

## A  Appendix

## B  Additional Experimental Details

In the following sections we provide additional experimental details required for reproduction of the experiments provided in the main text. All the experiments were conducted using Intel(R) Xeon(R) CPU E5-2620 v3 @2.4Ghz x2 (12 cores total).

### B.1  Selecting the Embedding Dimension

Choosing the embedding dimension might require application-specific considerations. For example, for clustering purposes, if the number of clusters is known ($K$), we might select $d = K$ to have a sufficient number of dimensions to separate the clusters. However, for other applications, we can use a validation set to tune the embedding dimension $d$; precisely, by increasing $d$ until the total correlation in the validation set stops increasing. Using this procedure we can estimate the most significant number of correlated embedding coordinates.

## B.2 Synthetic Example

For the linear model we use a learning rate of $0.005$ with $10,000$ epochs. The values of $\lambda^x$ and $\lambda^y$ are both set to $30$. These values were obtained using a cross validation procedure. The standard deviation $\sigma$ of the injected noise was set to $0.5$ by (Yamada et al., 2020). They have selected this value as it maximized the gradient of the regularization term at initialization. Empirically, we observe that for $\ell^0$-CCA smaller values of $\sigma$ translated to improved convergence. Specifically, we used $\sigma = 0.25$, which worked well in our experiments.

For each data model, we run the method 100 times with different realizations of $\boldsymbol{X}$ and $\boldsymbol{Y}$. In Table 1, we compare the average results of $\ell^0$-CCA to PMA (Witten et al., 2009), IP-SCCA (Mai & Zhang, 2019), SCCA-I (Hardoon & Shawe-Taylor, 2011),SCCA-II (Gao et al., 2017), and mod-SCCA (Suo et al., 2017). Results of SCCA-II, PMA, and mod-SCCA were simulated by (Suo et al., 2017).

To demonstrate the tuning procedure of $\lambda_x$ and $\lambda_y$, we generate data based on Model-I (described in Section 5.1) and evaluate the total correlation on training and validation samples. The results are presented in Fig. 7. As demonstrated in this plot, the correlation values on unseen samples can indicate the optimal values for the regularization parameters. Namely, the model leads to lower correlation values when selecting nuisance variables or removing some of the informative variables.

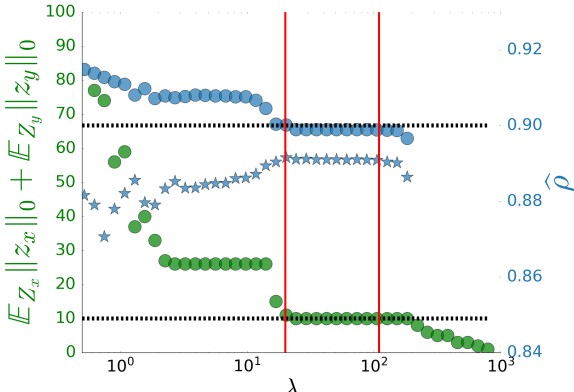

Figure 7: Regularization path of $\ell_0$-CCA on data generated from the linear model I (described in Section 5.1). Values on the left $y$-axis (green) represent the sum of active gates (by expectation). Values on the right $y$-axis (blue) represent the empirical correlation between the estimated representations, i.e. $\hat{\rho} = \hat{\boldsymbol{\phi}}^T \boldsymbol{X}^T \boldsymbol{Y} \hat{\boldsymbol{\eta}}$, where $\hat{\boldsymbol{\phi}}$ and $\hat{\boldsymbol{\eta}}$ are the estimated canonical vectors. Dashed lines indicate the correct number of active coefficients (10) and true correlation $\rho$ (0.9). In this example, we present the correlation values training samples (blue circles), and on unseen samples (blue asterisks). Red vertical lines indicate the range of $\lambda$'s that maximize the correlation on the validation set. Within this range of values the model selects the "correct" set of coefficients.

## B.3 Noisy MNIST

In this subsection we provide additional details regarding the noisy MNIST experiment. In Fig. 8, we present the performance as a function of the number of active gates (pixels) controlled by $\lambda^x = \lambda^y$. The Mutual Information (MI) score, $k$-means, and linear Support Vector Machine (LSVM) accuracies were computed based on $\ell^0$-DCCA embedding with learning rate of $0.01$. Furthermore, the number of epochs ($\sim 4000$) was tuned by early stopping using a validation set of size $10000$. To learn 10 dimensional correlated embedding, we use the same architecture as suggested by (Wang et al., 2015a) consisting of three hidden layers with $1000$ neurons each. The number of dimensions in the embedding was selected based on the number of classes in MNIST. This architecture is used for $\ell^0$-DCCA, DCCA (Andrew et al., 2013) and DCCAE (Wang et al., 2015b). Note that for $\ell^0$-DCCA using small values of the regularization parameters $\lambda^x$ and $\lambda^y$, increases the number of selected features and degrades the performance. This is duo to the fact that as more features are selected

more noise is introduced into the extracted representation (of size 10). It is interesting to note that the $k$-means was more robust to the introduced noise than the LSVM.

The regularization parameters $\lambda^x$ and $\lambda^y$ balance between the correlation loss and the amount of sparsification performed by the gates. These hyper parameters are tuned using the validation set by maximizing the total correlation value. We compare $\ell^0$-DCCA to CCA (Chaudhuri et al., 2009)[2], mod-SCCA (Suo et al., 2017), SCCA-HSIC (Uurtio et al., 2018), KCCA (Bach & Jordan, 2002)[3], NCCA (Michaeli et al., 2016) [4], multiview-ICA (Richard et al., 2020), DCCA (Andrew et al., 2013) [5] and DCCAE (Wang et al., 2015b). For all methods we use an embedding with dimension 10, and evaluate performance with $k$-means using 20 random initilizations, and using LSVM by performing training on the training samples and testing on the remaining samples (split defined in the main text). The hyperparameters of all methods are tuned to maximize the total correlation on a validation set. The solution obtained by mod-SCCA is based on $\sim 320$ variables. In this experiment we tried to train SCCA-HSIC (Uurtio et al., 2018) [6] for over two days, but it did not converge. Furthermore, we believe that the poor performance of the kernel methods stems from the high level of noise in the input images. We note that grad-KCCA (Uurtio et al., 2019) did not converge in this experiment, therefore we did not report its performance on MNIST.

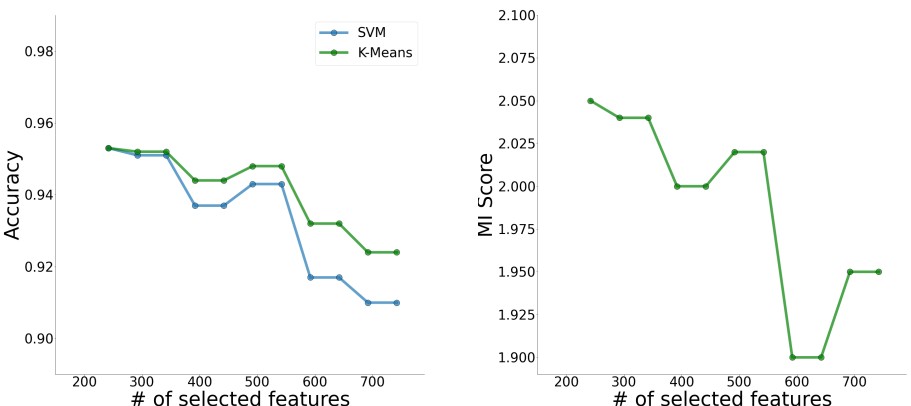

Figure 8: $k$-means and SVM classification accuracy (left) and mutual information score (right) vs. the number of selected features.

## B.4 SEISMIC EVENT CLASSIFICATION

Using the seismic data, we compare the performance of $\ell^0$-DCCA with a linear and non-linear activation. In this example, we use a learning rate of $0.01$ with $2000$ epochs. The number of neurons for the five hidden layer are: $300, 200, 100, 50$, and $40$ respectively, with a tanh activation after each layer. The number of dimensions in the embedding ($d = 3$) was selected based on the number of classes in the data. Parameters are optimized manually to maximize the correlation on a validation set. In Fig. 9, we present SVM accuracy for different levels of sparsity. The presented number of features is the average over both modalities, and SVM performance is evaluated using 5-folds cross validation. We compare $\ell^0$-DCCA to CCA (Chaudhuri et al., 2009), mod-SCCA (Suo et al., 2017), SCCA-HSIC (Uurtio et al., 2018), KCCA (Bach & Jordan, 2002), NCCA (Michaeli et al., 2016), multiview-ICA (Richard et al., 2020), DCCA (Andrew et al., 2013), and DCCAE (Wang et al., 2015b). For all methods we use an embedding with dimension $d = 3$, and evaluate performance with $k$-means using 20 random initilizations, and using linear SVM by performing a 5-folds cross validation. For the kernel methods we evaluated performance by constructing a kernel using $k = 5, 10, ..., 50$, nearest neighbors and selected the value which maximized performance

---

[2]https://scikit-learn.org/stable/modules/generated/sklearn.cross_decomposition.CCA.html

[3]https://gist.github.com/yuyay/16ce4914683da30f87d0

[4]https://tomer.net.technion.ac.il/files/2017/08/NCCAcode_v3.zip

[5]https://github.com/adrianna1211/DeepCCA_tensorflow

[6]https://github.com/aalto-ics-kepaco/scca-hsic

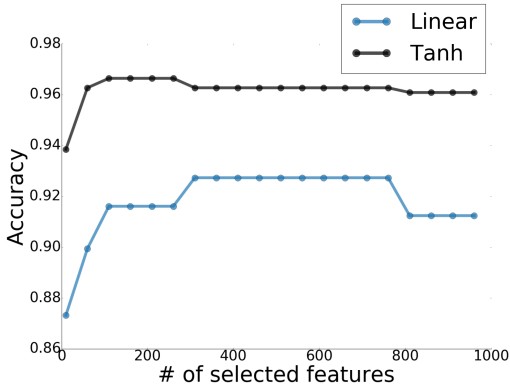

Figure 9: Classification accuracy on the noisy seismic data. Performance is evaluated using linear SVM in the 3 dimensional embedding. Comparing performance of $\ell^0$-DCCA for different levels of sparsity, and using linear and nonlinear activation (tanh).

in terms of total correlation. In this example, the optimal solution obtained mod-SCCA Suo et al. (2017) is based on $\sim 200$ input variables, while the solution obtained SCCA-HSIC Uurtio et al. (2018) is based on $\sim 77$ variables. This means that our method uses much fewer input variables and leads to substantially higher clustering and classification accuracies (see Table 3).

## C  LOW DIMENSIONAL SYNTHETIC EXAMPLE

Following an interesting suggestion raised during the review process, we provide here an evaluation of the method for the regime of $D \ll N$. We use a noisy version of the synthetic data model presented in Section 5.1. Specifically, we generate samples from $\boldsymbol{X}, \boldsymbol{Y} \in \mathbb{R}^{D \times N}$, using the following distribution $\begin{pmatrix} \boldsymbol{X} \\ \boldsymbol{Y} \end{pmatrix} \sim N\left( \begin{pmatrix} \mathbf{0} \\ \mathbf{0} \end{pmatrix}, \begin{pmatrix} \boldsymbol{\Sigma}_x & \boldsymbol{\Sigma}_{xy} \\ \boldsymbol{\Sigma}_{yx} & \boldsymbol{\Sigma}_y \end{pmatrix} \right)$, where $\boldsymbol{\Sigma}_{xy} = \rho_0 \boldsymbol{\Sigma}_x (\boldsymbol{\phi}\boldsymbol{\eta}^T)\boldsymbol{\Sigma}_y$. With $\boldsymbol{\Sigma} = \boldsymbol{I}_D$.

The we use additive noise to create the two modalities, namely $\tilde{\boldsymbol{X}} = \boldsymbol{X} + \boldsymbol{\epsilon}_X, \tilde{\boldsymbol{Y}} = \boldsymbol{Y} + \boldsymbol{\epsilon}_Y$. Where $\boldsymbol{\epsilon}_X$ and $\boldsymbol{\epsilon}_Y$ are drawn from $N(0, \boldsymbol{I}_D \sigma_N)$. We use $D = 20$ and $N = 4000$ to evaluate if our method is beneficial in the regime where there are more samples than variables. In table 3 we present the average errors for different noise levels (values of $\sigma_N$). As evident from this table, the proposed approach can provide a more accurate estimate of the sparse canonical vectors in the regime of $D \ll N$.

| | $(D = 20, N = 4000)$ | | |
| --- | --- | --- | --- |
| $\sigma_N$ | 0 | 1 | 2 |
| CCA | (0.0098,0.0099) | (0.034,0.036) | (0.256,0.258) |
| SCCA-II (Gao et al., 2017) | (0.0013,0.0008) | (0.0131,0.0097) | (0.6003,0.5881) |
| $\ell_0$-CCA | **(0.0005,0.0004)** | **(0.0065,0.0044)** | **(0.1075,0.0633)** |

Table 3: Evaluating the estimation quality of the canonical vectors $\boldsymbol{\phi}$ and $\boldsymbol{\eta}$. Each pair indicates $(e_{\boldsymbol{\phi}}, e_{\boldsymbol{\eta}})$, which are the estimation errors of $\boldsymbol{\phi}$ and $\boldsymbol{\eta}$ respectively. We compare the proposed $\ell_0$-CCA to standard CCA and to SCCA-II (Gao et al., 2017). Different columns indicate different additive noise levels.

## D  CANCER SUB-TYPE CLASSIFICATION

In the main text, we demonstrated that the embedding extracted with $\ell_0$-DCCA leads to more accurate cancer sub-type classification. Here, we provide additional details on the experiment described in Section 5.5. In this example, we use a learning rate of $0.5$ with $2000$ epochs. The number of neurons for the 3 hidden layers are: $500, 300, 100$, with a tanh activation after each layer. The number of dimensions in the embedding ($d = 10$) was selected based on the number of classes in

---

**Algorithm 1** $\ell_0$-DCCA

---

**Input:** Coupled data, $\{X, Y\}$, regularization parameters $\lambda^x, \lambda^y$, number of epochs $T$.
Initialize the gate parameters: $\mu_i^x = 0.5$ for $i = 1, \ldots, D^x$, and $\mu_i^y = 0.5$ for $i = 1, \ldots, D^y$.
**for** $t = 1$ **to** $T$ **do**
  Sample a stochastic gate (STG) vectors $\mathbf{z}^x, \mathbf{z}^y$ defined based on equation 4.
  Apply the STG to the data $\hat{X} = \mathbf{z}^x \odot X$ and $\hat{Y} = \mathbf{z}^y \odot Y$.
  Compute the loss $L = \mathbb{E}\big[-\bar{\rho}(\boldsymbol{f}(\hat{X}), \boldsymbol{g}(\hat{Y})) + \lambda^x \|\mathbf{z}^x\|_0 + \lambda^y \|\mathbf{z}^y\|_0\big]$, where the calculation of
  the total correlation $\bar{\rho}$ is based on Section 3.4. Where $\boldsymbol{f}(\hat{X}) = \boldsymbol{f}(\mathbf{z^x} \odot X | \boldsymbol{\theta^x}) \in \mathbb{R}^{d \times N}$, and
  $\boldsymbol{g}(\hat{Y}) = \boldsymbol{g}(\mathbf{z^y} \odot Y | \boldsymbol{\theta^y}) \in \mathbb{R}^{d \times N}$.
  Update $\boldsymbol{\theta}^x = \boldsymbol{\theta}^x - \gamma \nabla_{\boldsymbol{\theta}^x} L, \quad \boldsymbol{\theta}^y = \boldsymbol{\theta}^y - \gamma \nabla_{\boldsymbol{\theta}^y} L,$
    $\boldsymbol{\pi}^x = \boldsymbol{\pi}^x - \gamma \nabla_{\boldsymbol{\pi}^x} L, \quad \boldsymbol{\pi}^y = \boldsymbol{\pi}^y - \gamma \nabla_{\boldsymbol{\pi}^y} L.$
**end for**
Optional: apply linear CCA on $\boldsymbol{f}(\hat{X})$, and $\boldsymbol{g}(\hat{Y})$) to remove arbitrary scalings.

---

the data. Parameters are optimized manually to maximize the correlation on a validation set. We compare $\ell^0$-DCCA to CCA (Chaudhuri et al., 2009), mod-SCCA (Suo et al., 2017), SCCA-HSIC (Uurtio et al., 2018), KCCA (Bach & Jordan, 2002), NCCA (Michaeli et al., 2016), multiview-ICA (Richard et al., 2020), DCCA (Andrew et al., 2013), and DCCAE (Wang et al., 2015b). We use the same SVM and $k$-means scheme as described in Section B.4. The solution obtained by mod-SCCA is based on $\sim 940$ variables.

To provide additional intuition on why sparsity is useful in biomedical applications, we further analyze the results presented in Section 5.5 (the METABRIC dataset). In the experiment provided in this section, our method selects 169 genes from the RNA data and extracts an embedding that leads to improved identification (clustering and classification) of cancer subtypes. To understand if the variables selected by our model are interpretable, we analyze genes selected by the model. We have observed that out of the 169 selected genes, six are part of the known risk factors for breast cancer, namely the well-known PAM50 list. The PAM50 contains 50 genes known to be critical variables for classifying breast cancer subtypes Bernhardt et al. (2016). Our model has identified six genes (in an unsupervised fashion) out of the 42 possible PAM50 genes that were initially available in the METABRIC data. To demonstrate that this is a statistically significant finding, we compute the probability of selecting 169 genes out of a total of 15019 genes while having an overlap of 6 genes with another list of size 42. This probability is 5.39E-6 (the p-value), which indicates that this overlap is unlikely to have happened by chance. To conclude, our method was able to identify known markers for breast cancer subtypes classification without using any labels. Notably, one of these genes is 'ERBB2', which corresponds to the HER2/neu protein. 'ERBB2' is known to have a large copy number in patients classified to the HER2 cancer subtype. Thus the method found a strong association between the RNA and CNA of this gene.

# E PSEUDOCODE

In Algorithm 1 we provide a pseudocode description of the proposed approach.

# F RUN TIME ANALYSIS

Our method requires computing covariance matrices for $k_x$ and $k_y$ variables with N samples at each iteration. This scales like $\mathcal{O}(Nk_x^2 + Nk_y^2)$. To demonstrate the computational efficiency of our method, we run $\ell_0$-CCA for different values of $N$ and $D$ and evaluate the empirical computational complexity of the method. In Fig. 10 we present the average runtime over 100 runs, the data is generates following Model I from Section 5.1.

# G GENERALIZED $\ell_0$-BASED SPARSE CCA

The proposed $\ell_0$-CCA algorithm was designed and demonstrated on coupled datasets. However, in many applications, one may have access to multiple ($> 2$) datasets. There are many real-life

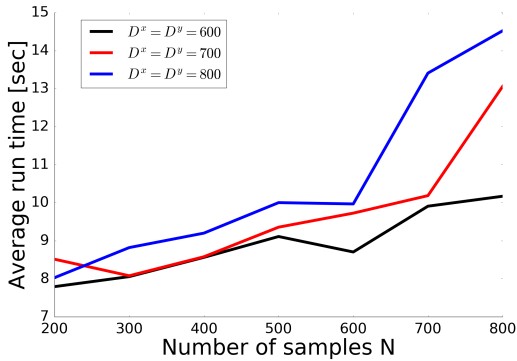

Figure 10: Run time evaluation of the proposed approach. We measure the average runtime (over 100 runs) for different values of $N$ and $D^x, D^y$.

cases where more than two modalities are available such as hyper-spectral imaging, medical imaging and more. The problem of CCA was generalized decades ago to include multiple data sets by (Horst, 1961; McKeon, 1966; McDonald, 1968) and (Carroll, 1968) to name some. These approaches were later summarized in (Kettering, 1971). Since then, many extensions have been proposed such as (Van de Geer, 1984). Some of the modern extensions of multi-view CCA comprise its regularised (Tenenhaus & Tenenhaus, 2011), kernelised (Tenenhaus et al., 2015) sparse (Tenenhaus et al., 2014) and deep (Benton et al., 2019) variants. In this section, we extend the proposed $\ell_0$-DCCA to multi-view datasets.

Let $K$ be the number of views and $\boldsymbol{X}^k$ the corresponding $k$-th view $1 \leq k \leq K$. The Deep Generalized CCA that was proposed in (Benton et al., 2019) aims to solve

$$\min_{\boldsymbol{G} \in \mathcal{R}^{n \times d}, \boldsymbol{U}_k \in \mathcal{R}^{d \times d}} \sum_{k=1}^{K} \|\boldsymbol{G} - \boldsymbol{U}_k^T f(\boldsymbol{X}^k)\|_F \quad s.t. \quad \boldsymbol{G}\boldsymbol{G}^T = \boldsymbol{I}, \tag{6}$$

where $\boldsymbol{G}$ is the desired shared representation, and $\boldsymbol{U}_k$ is a linear transformation from the $k$-th deep network output, $f(\boldsymbol{X}^k)$, to the shared representation space.

While the proposed method in Section 3.3 describes a solution for coupled views, this solution can be naturally extended to include multiple views by introducing a shared common space into the optimization problem in Eq 6. Denoting the random gating vectors $\boldsymbol{z}^k$ for view $\boldsymbol{X}^k$. The generalized $\ell_0$-DCCA loss is defined by

$$\min_{\boldsymbol{G} \in \mathcal{R}^{n \times d}, \boldsymbol{U}_k \in \mathcal{R}^{d \times d}, \boldsymbol{z}^k \in \mathcal{R}^{d_k}} \sum_{k=1}^{K} \|\boldsymbol{G} - \boldsymbol{U}_k^T f(\hat{\boldsymbol{X}}^k)\|_F + \lambda_k \|\boldsymbol{z}^k\|_0 \quad s.t. \quad \boldsymbol{G}\boldsymbol{G}^T = \boldsymbol{I}, \tag{7}$$

where $\boldsymbol{f}(\hat{\boldsymbol{X}^k}) = \boldsymbol{f}(\boldsymbol{z}^k \odot \boldsymbol{X}^k | \boldsymbol{\theta}^x) \in \mathbb{R}^{d \times N}$ and $\lambda_k$ is the regularization parameter that control the sparsity of $k$-th selected gate vector $\boldsymbol{z}^k$. The optimization problem in Eq 7 can be solved using standard optimizers such as gradient descent. The initialization of $\boldsymbol{G}$ and $\boldsymbol{U}_k$ plus the analysis of the above suggested generalized gated canonical correlation are left for further research.

## H    PROOF OF THEOREM 1

We first note that the equivalence of the solutions means that if we denote the solution to Eq. 2 by $(\hat{\boldsymbol{a}}, \hat{\boldsymbol{b}})$, then there is a corresponding set of values $\boldsymbol{\pi}^x, \boldsymbol{\pi}^y, \boldsymbol{\theta}^x, \boldsymbol{\theta}^y$, such that $(\hat{\boldsymbol{\alpha}}, \hat{\boldsymbol{\beta}}) = (\hat{\boldsymbol{a}}, \hat{\boldsymbol{b}})$, where $\boldsymbol{\alpha} = \boldsymbol{\theta}^x \odot \boldsymbol{s}^x$ and $\boldsymbol{\beta} = \boldsymbol{\theta}^y \odot \boldsymbol{s}^y$. The proof follows the same construction as the proof of Theorem 1 in (Yin et al., 2020). To prove the Theorem, we will show that the optimal solution to the deterministic sparse CCA problem (in Eq. 2) is a valid solution to the probabilistic sparse CCA problem (in Eq. 3) and vice versa.

First, we denote by $(\hat{a}, \hat{b})$ the optimal solution of the problem defined in Eq. 2, then by setting $\hat{\alpha} = \hat{a} \odot \hat{s}^x$ and $\hat{\beta} = \hat{b} \odot \hat{s}^y$, where $\hat{\pi}_i^x = 1$ if $\hat{a}_i \neq 0$ and $\pi_i^x = 0$ otherwise, we get a feasible solution to Eq. 3 which leads to the same objective value.

Now, we want to show that the optimal solution to Eq. 3 is also a feasible solution to Eq. 2. Denoting the optimal solution to Eq. 3 by $\hat{\alpha}$ and $\hat{\beta}$, with $\hat{\alpha} = \hat{\theta}^x \odot \hat{s}^x$ and $\hat{\beta} = \hat{\theta}^y \odot \hat{s}^y$, if $p(s^x|\pi^x)$ or $p(s^y|\pi^y)$ are point mass densities (i.e., $\pi^x, \pi^y \in \{0,1\}^D$), then the solution is effectively deterministic and $\hat{a} = \hat{\alpha}, \hat{b} = \hat{\beta}$ would be a valid solution to problem Eq. 2. If $p(s^x|\pi^x)$ and $p(s^y|\pi^y)$ are not point mass densities, we will now show by contradiction that all the points in the support of $p(s^x|\pi^x)$ (or $p(s^y|\pi^y)$) would lead to the same objective values. Assume without loss of generality that $\{s_1^x, ...., s_L^x\} \in \text{supp}[p(s^x|\hat{\pi}^x)]$, if not all values in the support lead to the same objective value, this means that there exist $s_l^x, s_m^x$ such that $L(s_l^x) < L(s_m^x)$, where

$$L(s^x) = \left[ - \rho((\hat{\theta}^x \odot s^x)^T X, (\hat{\theta}^y \odot \hat{s}^y)^T Y) + \lambda^x \|s^x\|_0 + \lambda^y \|\hat{s}^y\|_0 \right],$$

which is the objective from Eq. 3 with optimal values of $\hat{\theta}^x, \hat{\theta}^y, \hat{s}^y$. Now if we set $\bar{\pi}^x = s_l^x$ we would obtain $\mathbb{E}_{s^x \sim p(s^x|\bar{\pi}^x)} L(s^x) < \mathbb{E}_{s^x \sim p(s^x|\hat{\pi}^x)} L(s^x)$ which contradicts the assumption that $\hat{\pi}^x$ is optimal. This means that all points in $\text{supp}[p(s^x|\hat{\pi}^x)]$ lead to the same objective value, therefore, they are also feasible solutions to Eq. 2. Now, showing that the solution to Eq. 3 is a feasible solution to Eq. 2 and vice versa completes our proof.

# I LIMITATIONS AND FUTURE DIRECTIONS

The proposed method provides an effective solution to the problems of sparse linear and nonlinear CCA. One advantage of the suggested method compared with existing sparse CCA solutions, is that it can embed data into a $d > 1$ dimensional space and learn the sparsity pattern simultaneously. The proposed $\ell_0$-CCA problem albeit not being convex leads to an empirically stable solution across different settings. Nonetheless, the method has some limitations, specifically, tuning $\lambda$ requires a cross validation procedure, which could be costly in high dimensional regime. Another caveat of the existing approach is that it lacks guarantees when trained on small batches. In the future, we aim to extend the method to enable compatibility with small batch training. Furthermore, we are currently working on evaluating how the estimation error scales with respect to the dimension, sparsity, for different structures of the covariance matrix. Furthermore, we are working on bounding the estimation error of our method. We observe that using a Gaussian based data model (similar to the one used in section 5.1), it is possible to bound the estimation error (in probability) by $\mathcal{O}(D/N + ck/N)$ where $D$ is the number of variables, $N$ is the number of measurements, k is the true sparsity of the conical vectors, and $c$ is an independent constant. Although this bound provides some guarantee, it is ubiquitous in the context of the regime studied in our manuscript (i.e., $N \ll D$). Therefore, we are currently working on improving this bound. Empirically, we observe that the method works well in regimes that existing $\ell_1$ and $\ell_0$ based approaches fail.

