# OpenReview forum: "L0-Sparse Canonical Correlation Analysis"
_ICLR.cc/2022/Conference — ICLR 2022 Poster_

### Official Review · Reviewer_LbVS · 2021-11-01

**Correctness:** 3
**Technical Novelty And Significance:** 3
**Empirical Novelty And Significance:** 3
**Recommendation:** 6
**Confidence:** 4

**Main Review:**

Comments:

Originality & Quality: The paper proposes a stochastic gating based approach for l0-CCA which uses ideas from a couple of recent papers on gaussian relaxation of discrete/bernoulli random variables. The resulting algorithm is straightforward and it is also possible to "backprop into it", so it is also amenable to non-linear "deep" extensions. The results on various synthetic and real-world datasets show the superior performance of the proposed method.

While I like the proposed approach, there are a couple of questions/concerns that I have:

1). The paper mentions that standard l0-CCA relies on greedy optimization and that the l1-CCA leads to parameter shrinkage. Both these are valid concerns. Though, it is unclear what recovery guarantees does the stochastic gating approach has? Optimal subset selection is a NP-Hard problem, so clearly the proposed approach is also performing some approximation. Can it be theoretically compared to the greedy l0-solution?

2). The paper shows predictive accuracies of the different CCA approaches on real-world datasets. However, it doesn't show the number of features selected by the different methods and how they compare to each other. If we only cared about predictive accuracy then why select features as shown by (Shalev-Shwartz et al. 2010)? And, also sparsity leads to interpretability so it is unclear how different approaches compare on that front.



Clarity: The paper is well written and puts itself nicely in context of previous work. The overall presentation of the paper is good.

Significance: The paper addresses an important problem of sparse CCA using a l0 penalty. The proposed approach uses stochastic gating to enforce sparsity which does not rely on a greedy/heuristic solution as previous l0-sparse CCA approaches.

**Summary Of The Paper:**

The paper proposes a new method for l0-CCA using stochastic gating which allows for an efficient algorithm and also permits a deep-version of the l0-CCA. Results are shown on synthetic and real-world datasets that highlight the superior performance of the proposed approach.

Main Contributions:

1). The paper proposes a new approach for l0-CCA that uses a continuous relaxation of a bernoulli random variable via stochastic gating.

2). The paper extends the proposed method to l0-deep-CCA which allows non-linear interactions between the two CCA views.

3). Results are shown on synthetic and real-world datasets which show the superior performance of the proposed l0-CCA methods.

**Summary Of The Review:**

An interesting paper that solves an "old" problem of sparse-CCA using some recent advances in stochastic gating based sparsity solution. Results are comprehensive, but the focus is less on feature selection and more on predictive power. If predictive power is the primary concern, why care about sparsity?

---

> ### Author Response · Authors · 2021-11-19
> **Reply to reviewer LbVs (part 1)**
>
> We thank the reviewer for pointing out the strengths and weaknesses of the proposed approach. In response to the constructive suggestions provided by all four reviewers, we have spent the past week conducting new experiments and revising our manuscript to incorporate the new information. In the following paragraph, we address the questions raised by the reviewer.
> Q) Though, it is unclear what recovery guarantees does the stochastic gating approach has? Optimal subset selection is a NP-Hard problem, so clearly the proposed approach is also performing some approximation. Can it be theoretically compared to the greedy l0-solution?
>
> A) Currently, there are no theoretical guarantees for the greedy l0 approach. In contrast, there are several minimax rates for the l1 based schemes (see [1]). The proposed solution is nonconvex, which makes the analysis harder; however, using a Gaussian based data model (similar to the one used in section 5.1), it is possible to bound the estimation error (in probability) by ~ D/N +ck/N where D is the number of variables, N is the number of measurements, k is the true sparsity of the canonical vectors, and c is an independent constant. This is based on preliminary results (available as a technical report on arxiv) that we prefer not to cite here to avoid compromise of anonymity. A discussion about this point already appears in Appendix Section F. Although this bound provides some guarantee, it is impractical in the context of the regime studied in our manuscript (i.e., D>N). Therefore, we are currently working on improving this bound. Empirically, we observe that the method works well in regimes that existing L1 and L0 based approaches fail.
>
> [1] Gao, C., Ma, Z., Ren, Z., & Zhou, H. H. (2015). Minimax estimation in sparse canonical correlation analysis. The Annals of Statistics, 43(5), 2168-2197.
>
> Q) If we only cared about predictive accuracy then why select features as shown by (Shalev-Shwartz et al. 2010)? And, also sparsity leads to interpretability so it is unclear how different approaches compare on that front.
>
> A) In specific settings for supervised learning, as Shalev-Shwartz et al., there is a tradeoff between sparsity and the model's accuracy. However, in the context of CCA, when the number of variables is larger than the number of samples, higher correlation often results from overfitting and does not lead to better generalization. Moreover, if the data consist of nuisance variables, sparsity can reduce the generalization gap even if N>D. These properties are demonstrated empirically in sections 5.3-5.5. As the reviewer noted, sparsity also improves interpretability. To confirm that the selected features are interpretable, we have presented the features chosen in two out of the three real examples included in the paper. In the MNIST example, the method correctly separates center pixels from background pixels (see Figure 5). In the Seismic experiment, the method identifies ~17 time-frequency bins (variables) that coincide with the expert knowledge about the properties of the Primary and Secondary waves generated by the seismic event (see Figure 6). In comparison, the optimal solution obtained mod-SCCA [12] is based on ~ 200 input variables, while the solution of SCCA-HSIC [20] is based on ~77 variables. This means that our method uses much fewer input variables and leads to substantially higher clustering and classification accuracies (see Table 3).  We have added the number of selected features by different baselines to the manuscript.

---

> > ### Author Response · Authors · 2021-11-19
> > **Reply to reviewer LbVs (part 2)**
> >
> > Q)Results are comprehensive, but the focus is less on feature selection and more on predictive power. If predictive power is the primary concern, why care about sparsity?
> >
> > A) We agree with the reviewer that the method is essential for feature selection and improving predictive power. In the paper, we have demonstrated the capabilities of the technique in the context of both tasks. For improving predictive power, the method was compared to several baselines demonstrating that it leads to a reduced representation that is more reliable to the classes in the data (see Table 2). In the context of feature selection, we have first used synthetic examples, in which the method outperformed other sparse models in its ability to recover the correct sparse canonical vectors. Our model achieved low values in terms of estimation error; this indicates that it correctly recovered the indices of the active features in the synthetic linear example.
> > Other real examples evaluated in the paper also indicate the ability of the method to recover informative correlated variables based on high dimensional observations. Specifically, in the MNIST, two-puppets, and Seismic examples, the figures of the selected features indicate qualitatively that our model can identify the correlated variables in these datasets.
> >
> > To provide additional intuition on why sparsity is useful in biomedical applications, we further analyze the results presented in section 5.5 (the METABRIC dataset). In the experiment provided in this section, our method selects 169 genes from the RNA data and extracts an embedding that leads to improved identification (clustering and classification) of cancer subtypes. To understand if the variables selected by our model are interpretable, we analyze genes selected by the model. We have observed that out of the 169 selected genes, six are part of the known risk factors for breast cancer, namely the well-known PAM50 list. The PAM50 contains 50 genes known to be critical variables for classifying breast cancer subtypes [2]. Our model has identified six genes (in an unsupervised fashion) out of the 42 possible PAM50 genes that were initially available in the METABRIC data. To demonstrate that this is a statistically significant finding, we compute the probability of selecting 169 genes out of a total of 15019 genes while having an overlap of 6 genes with another list of size 42. This probability is 5.39E-6 (the p-value), which indicates that this overlap is unlikely to have happened by chance.
> >
> > To conclude, our method was able to identify known markers for breast cancer subtypes classification without using any labels. Notably, one of these genes is ‘ERBB2’, which corresponds to the HER2/neu protein. ‘ERBB2’ is known to have a large copy number in patients classified to the HER2 cancer subtype. Thus the method found a strong association between the RNA and CNA of this gene.
> > [2] Sarah Bernhardt et al. Hormonal modulation of breast cancer gene expression: implications for intrinsic subtyping in premenopausal women, 2016.

---

> > > ### Comment · Reviewer_LbVS · 2021-11-22
> > > **Thank you for your detailed responses.**
> > >
> > > Dear authors,
> > >
> > > Thanks for your comprehensive responses to my concerns! I am satisfied with your answers.

---

### Official Review · Reviewer_snhT · 2021-11-01

**Correctness:** 4
**Technical Novelty And Significance:** 3
**Empirical Novelty And Significance:** Not applicable
**Recommendation:** 6
**Confidence:** 5

**Main Review:**

It is very critical to select relevant features from high dimensional multimodal data and such method is very demanding in lots of real cases.

One key strength of the presented method is that it is able to sub-select relevant features from high dimensional inputs and it showed better performance compared to others for predictive purpose.

The paper is well written, and the overall presentation of the paper is good.

Questions:

1. How to select the most relevant features in the multimodal learning field is crucial for model interpretation and downstream analysis and it will have wide applications in biology or human genetics. For those fields, researchers are more interested to select which gene, protein, or biomarkers are most important for disease development. Even though the paper showed it’s outperformed predictive performance, however, it missed the detailed model interpretation part, feature selection. I would be interested to see one biology dataset application for model interpretability on this method.

2. In figure 2, the wide range of lambda showed around 40 coefficients. Please comment on how could the algorithm avoid the wrong number of coefficients?

3. How does the proposed method compare to other methods regarding accuracy vs sparsity?



**Summary Of The Paper:**

 I have seen this manuscript before. The main contribution of this paper is to combine two existing works, Gaussian-based continuous relaxation of Bernoulli random variables, gates (Yamada et al., 2020), and sparse CCA (Suo et al., 2017).

**Summary Of The Review:**

I like the paper overall. However, I think the model interpretation is more important than the prediction for multimodal learning/CCA, specifically, the key strength of this proposed paper is feature selection.

---

> ### Author Response · Authors · 2021-11-17
> **Reply to reviewer snhT**
>
> Reply: We thank the reviewer for pointing out his appreciation of the paper and its practical importance to multimodal data analysis. In response to the constructive suggestions provided by all four reviewers, we have spent the past week conducting new experiments, and revising our manuscript to incorporate the new information. In the following paragraph, we clarify the concerns raised by the reviewer.
>
> Q) I would be interested to see one biology dataset application for model interpretability on this method.
>
> A) Unsupervised selection of informative variables in biological datasets is essential for studying different medical conditions. Following the reviewer's suggestion, we provide here an additional analysis of the results presented in section 5.5 (the METABRIC dataset). In the experiment provided in this section, our method selects 169 genes from the RNA data and extracts an embedding that leads to improved identification (clustering and classification) of cancer subtypes. To understand if the variables selected by our model are interpretable, we analyze genes selected by the model. We have observed that out of the 169 selected genes, six are part of the known risk factors for breast cancer, namely the well known PAM50 list. The PAM50 contains 50 genes known to be critical variables for classifying breast cancer subtypes [1]. Our model has identified six genes (in an unsupervised fashion) out of the 42 possible PAM50 genes that were initially available in the METABRIC data. To demonstrate that this is a statistically significant finding, we compute the probability of selecting 169 genes out of a total of 15019 genes while having an overlap of 6 genes with another list of size 42. This probability is 5.39E-6 (the p-value), which indicates that this overlap is unlikely to have happened by chance. To conclude, our method was able to identify known markers for breast cancer subtypes classification without using any labels. Notably, one of these genes is ‘ERBB2’ which corresponds to the HER2/neu protein. ‘ERBB2’ is known to have a large copy number in patients classified to the HER2 cancer subtype. Thus the method found strong association between the RNA and CNA of this gene.
>
> In other examples our model also leads to interpretable selection of variables. For instance in the MNIST example, the method correctly separates center pixels from background pixels (see Figure 5). In the Seismic experiment, the method identifies ~17 time-frequency bins (variables) that coincide with the expert knowledge about the properties of the Primary and Secondary waves generated by the seismic event (see Figure 6).
> [1] Sarah Bernhardt et al. Hormonal Modulation of Breast Cancer Gene Expression: Implications for Intrinsic Subtyping in Premenopausal Women, 2016.
>
> Q) In figure 2, the wide range of lambda showed around 40 coefficients. Please comment on how could the algorithm avoid the wrong number of coefficients?
>
> A) We use a cross-validation procedure to tune lambda in an unsupervised fashion to select the “correct” number of variables. Specifically, we can avoid selecting the “wrong” number of coefficients by leaving a validation set and selecting lambda so that the total correlation on the validation set is maximized. We will add a plot of the correlation on the validation set to the revised manuscript to demonstrate this procedure. In all examples evaluated in section 5.1, the maximal correlation on the validation set is attained when the model selects the correct number of coefficients.
>
> Q)How does the proposed method compare to other methods regarding accuracy vs sparsity?
>
> A) As demonstrated in sections 5.3-5.5, sparsity can often lead to improved classification or clustering accuracy when evaluating the method on real data. This happens if the data consist of nuisance variables (i.e., information poor or noisy). In the Seismic experiment, the L0-DCCA identifies 17 time-frequency bins (variables) that coincide with the expert knowledge about the properties of the Primary and Secondary waves generated by the seismic event (see Figure 6). In comparison, the optimal solution obtained mod-SCCA [12] is based on 200 input variables, while the solution obtained SCCA-HSIC [20] is based on 77 variables. While all methods sparsify nuisance features, our approach uses much fewer input variables and leads to substantially higher clustering and classification accuracies (see Table 3). We have added the number of features selected by other baselines to table 2. Specifically, mod-SCCA selects 320 and 941 features for MNIST and the METABRIC data, while SCCA-HSIC did not converge on these datasets.

---

> > ### Comment · Reviewer_snhT · 2021-11-28
> > **Thanks for your response**
> >
> > Dear authors,
> >
> > Thanks for your responses to my concerns and I am satisfied with your answers.

---

### Official Review · Reviewer_R5ad · 2021-11-03

**Correctness:** 3
**Technical Novelty And Significance:** 3
**Empirical Novelty And Significance:** 3
**Recommendation:** 6
**Confidence:** 4

**Details Of Ethics Concerns:**

No ethics concerns.

**Main Review:**

This paper is technically sound and interesting, in that the authors demonstrate how sparse CCA with l_0 norm can be effectively solved, and how the sparsity constraint can be integrated into the deep CCA model. My concerns lie mainly in the experimental section.

In the supplementary material, the authors mention that the parameters are optimized to maximize the correlation for the cancer classification task. Why don’t choose parameters based on accuracy instead of correlation? Does high correlation guarantee better accuracy or vice versa? Are the parameters of all compared methods chosen in the same manner?

2. How are lambda_x and lambda_y selected since they directly control the sparsity of variables?

3. The authors claim that the proposed method is efficient in C.2 (Run Time Analysis), it seems that the model scales when k increases. Besides, training a DCCA model for classification seems to be time-consuming. Therefore, it is also important to compare the time complexity and running time (or training time) of compared methods.

4.  In Table 2, why don’t the authors include the results of the proposed sparse CCA method?

5. The datasets included are mainly for vector-based applications. Recently, there are several tensor-based CCA methods, it is interesting to also report results on some high-dimensional datasets, for example, Gait and JAFFE.

**Summary Of The Paper:**

The paper proposes a sparse canonical correlation analysis method based on l_0 norm. A continuous relaxation scheme is adopted for solving sparse CCA. The proposed model is then extended to nonlinear function estimation and combined with deep neural networks. Experimental results on synthetic and real-world datasets demonstrate the effectiveness of the proposed methods.

**Summary Of The Review:**

The paper is technically sound, but the authors should carefully revise their experimental section to address my concerns.

---

> ### Author Response · Authors · 2021-11-17
> **Response to Reviewer R5ad**
>
> We thank the reviewer for pointing out the strengths and weaknesses of our proposed approach. In response to the constructive suggestions provided by all four reviewers, we have spent the past week conducting new experiments, and revising our manuscript to incorporate the new information. In the following paragraph, we clarify the concerns raised by the reviewer. Our response was also integrated into the manuscript to improve the readability of the paper.
>
> Q)Why don’t choose parameters based on accuracy instead of correlation? Does high correlation guarantee better accuracy or vice versa?
>
> A) Hyperparameter tuning-  In the paper, we evaluated the proposed approach in an unsupervised fashion. Namely, we tuned $\lambda$ to maximize the proposed objective while avoiding overfitting as much as possible. This could be achieved by selecting $\lambda$ that maximizes the correlation on unseen samples (validation set). We completely agree with the reviewer that there is no guarantee that this is the optimal choice for classification. If labels exist, they could be used to tune the parameters further. Nonetheless, as demonstrated empirically in the paper, this unsupervised tuning procedure leads to a representation that faithfully preserves the class information, and improves classification and clustering results compared to other dimensionality reduction methods. This tuning procedure also works well on the synthetic examples evaluated in Section 5.1.
>
> Q)Are the parameters of all compared methods chosen in the same manner?
>
> A) Yes, all parameters of all compared methods are chosen in the same manner. Namely, to maximize the objectives of these methods on unseen samples.
>
> Q)How are lambda_x and lambda_y selected since they directly control the sparsity of variables?
>
> A) lambda_x and lambda_y are tuned to maximize the total correlation on unseen samples (a validation set). Sparsification of the variables helps by removing modality-specific nuisance variables that may lead to spurious correlations on the training samples.
>
> Q)Therefore, it is also important to compare the time complexity and running time (or training time) of compared methods.
>
> A) This is a great suggestion; we will incorporate a runtime comparison to other baselines in the revised manuscript.
>
> Q)In Table 2, why don’t the authors include the results of the proposed sparse CCA method?
>
> A) Since the real data model may be nonlinear, we restricted our evaluation of the proposed approach to nonlinear L0-DCCA.
> Nonetheless, following this suggestion, we will add the performance of the linear variant of our approach (L0-CCA) to table 2.
>
> Q) It is interesting to also report results on some high-dimensional datasets, for example, Gait and JAFFE.
>
> A) We performed a lengthy online search and were not able to find any paper or dataset by Gait and JAFFE. Could the reviewer please point out the specific reference?

---

> > ### Comment · Reviewer_R5ad · 2021-11-29
> > **Thanks for your response**
> >
> > Dear authors,
> >
> > Thanks for your responses to my questions. I am satisfied with your answers and have no further questions.

---

### Official Review · Reviewer_N6Rd · 2021-11-07

**Correctness:** 3
**Technical Novelty And Significance:** 3
**Empirical Novelty And Significance:** 3
**Recommendation:** 6
**Confidence:** 3

**Details Of Ethics Concerns:**

Authors present limitations do address limitations of proposed approach but would like to see performance for N>>D for sanity check as well. There are no direct social impact issues or concerns.

**Main Review:**

Idea of sparse non-linear CCA for N << D case is novel and interesting and is valuable for such real-world cases. Loss function that minimizes correlation and increases sparsity provides a unique way to do feature selection for multi-modal cases.

Algorithm presented that does l_0 feature selection using Gaussian relaxed Bernoulli variables is interesting and novel application of for sparse-DCCA problem.

Empirical results on synthetic data show lowest error in learned correlations from sparse-DCCA compared to other SOTA techniques. Results on multi-view, MNIST digits, Seismic event and METABRIC datasets show that technique is able to pick relevant input features in challenging noisy and high dimensional (N<<D) cases with few examples.

A strength of the presented method is that it is able to sub-select relevant features from high dimensional inputs so it is expected to perform well compared to techniques that are not specifically designed for that. It would be helpful to see performance of sparse-DCCA for cases of N>>D compared to other techniques for comparison.

**Summary Of The Paper:**

Paper presents a novel sparse nonlinear-CCA method which works especially for cases where number of sample are less than max input feature dimension (D) N << D. It presents a novel algorithm that uses Gaussian relaxation of Bernoulli gating variables that sub-select sparse set of features. Empirical results on synthetic and real world data show efficacy of the method.

**Summary Of The Review:**

Novelty of the paper is extension of idea of nonlinear-CCA method to sparse case where where number of sample are less than max input feature dimensions (D) N << D. Results presented are convincing and support technical details.

---

> ### Author Response · Authors · 2021-11-17
> **Reply to Reviewer N6Rd**
>
> We thank the reviewer for acknowledging the novelty of our work and appreciating its practical importance. In response to the constructive suggestions provided by all four reviewers, we have spent the past week conducting new experiments, and revising our manuscript to incorporate the new information. Here, we address the reviewer’s main question.
>
> Q) It would be helpful to see performance of sparse-DCCA for cases of N>>D compared to other techniques for comparison.
>
>
> A) The reviewer raises an interesting question regarding the performance of the method for N>>D. In the paper, we demonstrated that the method works well on MNIST, which is in the regime of N>D. Specifically, N=60,000 and D=784, the method extracted a 10-dimensional embedding that improved classification and clustering accuracy compared to several baselines. Following the reviewer's suggestion, we performed a new experiment to evaluate if the method is beneficial for N>>D cases. To this end, we generate data based on model-I (see Section 5.1), with D=20, N=4000. In this regime, CCA is expected to work well. Nonetheless, we observe that our model improves the recovery of the canonical vectors compared to CCA. Specifically, the average recovery errors $e_{{\phi}}=2(1-|{\phi}^T\hat{{\phi}}|)$ are 0.0098, 0.0013, and 0.0005 for CCA, SCCA (Gao et al.) and L0-CCA (proposed) respectively. This indicates that our method can lead to a more reliable recovery of the sparse canonical vectors even when N>>D. Next, we explore a more challenging setting by adding modality-specific Gaussian noise to X and Y drawn from a Gaussian distribution with zero mean and standard deviation of 2. In this setting the average recovery errors $e_{{\phi}}=2(1-|{\phi}^T\hat{{\phi}}|)$ are 0.258, 0.58, and 0.0063 for CCA, SCCA (Gao et al.) and L0-CCA (proposed) respectively. This suggests that our model can lead to a substantial performance gain compared to other baselines in the noisy setting.

---

### Author Response · Authors · 2021-11-22
**Description of changes made in the paper**

We thank all reviewers for spending valuable time reading our paper and for providing helpful comments and suggestions. We appreciate that the reviewers acknowledge the importance of the problem and the novelty of our method. In this response, we detail the changes made in the paper following the suggestions made by the reviewers.

1) Evaluation of the method for $D\ll N$ (a suggestion made by Reviewer N6Rd):
We have added a new set of synthetic experiments to Appendix Section C.

2) Hyperparameter tuning procedure (questions raised by Reviewer R5ad and snhT):
The tuning procedure was originally described in the main text and Appendix. Nonetheless, to demonstrate the procedure, we added a new synthetic example to Appendix Section B.2. In the new example, also presented in Figure 7, we show that the correlation values on unseen samples (validation set) can be used to tune the method's hyperparameters properly.

3) Performance of $\ell_0$-CCA on real data (question raised by Reviewer R5ad):
Following the suggestion raised by the reviewer, we have applied the linear variant of our method $\ell_0$-CCA to all real datasets evaluated in the paper. The new results were added to Table 2 in the revised manuscript. The results of this experiment indicate that $\ell_0$-CCA leads to substantial improvements compared to other linear models (PCA, CCA, mod-SCCA). Furthermore, since our model sparsifies the input space, it can perform at par with other non-linear models on the Seismic and METABRIC data.

4) The sparsity of other methods on real data (Question raised by Reviewer snhT):
We have added the details about the number of selected features by other sparse CCA models to Appendix Sections B.3, B.4, and D.

5) Interpretability of our feature selection on biological data (question raised by Reviewer LvBs):
We have added a new paragraph to Appendix Section D. In this paragraph; we analyze the genes selected by our model on the METABRIC data.

---

### Decision · Program_Chairs · 2022-01-20

**Decision:**

Accept (Poster)

**Comment:**

Canonical correlation analysis is a method for studying associations between two sets of variables. However these methods lose their effectiveness when the number of variables is larger than the number of samples. This paper proposes a method, based on stochastic gating, for solving a $\ell_0$-CCA problem where the goal is to learn correlated representations based on sparse subsets of variables. Essentially, this paper combines ideas from Yamada et al. and Suo et al. who introduced Gaussian-based relaxations of Bernoulli random variables, and sparse CCA respectively. They also extend their methods to work with nonlinear functions by integrating deep neural networks into the $\ell_0$-CCA model. They gave experimental results on various synthetic and real examples, including to feature selection on biological data. The author response addressed a number of the reviewers' concerns, including by providing additional experiments and analyzing the genes selected by their model on the METABRIC dataset. Overall this is a solid contribution both from a theoretical and experimental standpoint.